# Effects of Radio Frequency Tempering on the Temperature Distribution and Physiochemical Properties of Salmon (*Salmo salar*)

**DOI:** 10.3390/foods11060893

**Published:** 2022-03-21

**Authors:** Rong Han, Jialing He, Yixuan Chen, Feng Li, Hu Shi, Yang Jiao

**Affiliations:** 1College of Food Science and Technology, Shanghai Ocean University, Shanghai 201306, China; sxczhr@163.com (R.H.); m13974501128_1@163.com (J.H.); yxchen@shou.edu.cn (Y.C.); fli@shou.edu.cn (F.L.); hshi@shou.edu.cn (H.S.); 2Engineering Research Center of Food Thermal-Processing Technology, Shanghai 201306, China

**Keywords:** tempering, radio frequency, quality, salmon (*Salmo salar*), fish

## Abstract

Salmon (*Salmo salar*) is a precious fish with high nutritional value, which is perishable when subjected to improper tempering processes before consumption. In traditional air and water tempering, the medium temperature of 10 °C is commonly used to guarantee a reasonable tempering time and product quality. Radio frequency tempering (RT) is a dielectric heating method, which has the advantage of uniform heating to ensure meat quality. The effects of radio frequency tempering (RT, 40.68 MHz, 400 W), water tempering (WT + 10 °C, 10 ± 0.5 °C), and air tempering (AT + 10 °C, 10 ± 1 °C) on the physiochemical properties of salmon fillets were investigated in this study. The quality of salmon fillets was evaluated in terms of drip loss, cooking loss, color, water migration and texture properties. Results showed that all tempering methods affected salmon fillet quality. The tempering times of WT + 10 °C and AT + 10 °C were 3.0 and 12.8 times longer than that of RT, respectively. AT + 10 °C produced the most uniform temperature distribution, followed by WT + 10 °C and RT. The amount of immobile water shifting to free water after WT + 10 °C was higher than that of RT and AT + 10 °C, which was in consistent with the drip and cooking loss. The spaces between the intercellular fibers increased significantly after WT + 10 °C compared to those of RT and AT + 10 °C. The results demonstrated that RT was an alternative novel salmon tempering method, which was fast and relatively uniform with a high quality retention rate. It could be applied to frozen salmon fillets after receiving from overseas catches, which need temperature elevation for further cutting or consumption.

## 1. Introduction

Salmon (*Salmo salar*) is a good source of n-3 polyunsaturated fatty acids, essential amino acids and bioactive peptides [1,2], which makes it a great supplement to people’s daily diets. The consumption of salmon has increased gradually thanks to its characteristics of high nutrition and delicious taste. However, fish and fish products are susceptible to spoilage and deterioration; thus, their shelf-lives are usually short. Freezing is a convenient and effective method for fish preservation, while tempering is usually a necessary process for the further processing, marketing, and cooking of frozen fish while restoring its quality. Improper tempering methods, which usually result in a longer tempering time and uncontrollable temperature, may cause physical and chemical damages and changes in the fish, such as higher drip loss [3,4], color change [5,6], protein denaturation and lipid oxidation [6,7,8,9,10] and texture change [11]. Conventional tempering processes currently applied in the food industry, including air and water tempering, were characterized as time-consuming and causes of food quality losses, while new tempering technologies may present the advantages of a fast tempering rate, energy saving and good quality [12]. Thus, more novel tempering technologies have been emerging and studied to replace the traditional technologies.

Radio frequencies (RF) are electromagnetic waves with a frequency range of 3 kHz to 300 MHz, and the commonly used frequencies in the food industry are 13.56, 27.12, and 40.68 MHz [13,14,15,16]. RF heating is a dielectric heating method with the characteristics of a high heating rate, volumetric heating, uniform heating, and a controllable process to ensure food quality and hygiene [17,18]. The RF generator produces an alternating electromagnetic field, and when the food is exposed to this alternating electromagnetic field, the dipole molecules, atoms, and ions in the food material rotate and rearrange, causing friction that generates heat inside the food [19]. Due to the longer wavelength compared to microwave, RF technology is more suitable for heating large volume samples and minimizing non-uniform heating because of its relatively higher penetration depth [20].

RF tempering of food products has been researched for its application to various products in recent years. Farag et al. [21] thawed lean beef using RF and discovered that RF could reduce the drip and micronutrient losses during the tempering process compared with conventional tempering methods. It was also found that the tempering rate of RF was 85 folds reduced compared to air tempering. Llave et al. [22] discovered the RF tempering rate of tuna fish was three times higher than that with air tempering and that the best uniformity occurred with a top electrode projection similar in size to the sample. Kim et al. [23] investigated RF tempering of pork with a curved-shape top electrode and found that temperature uniformity was improved by using a top electrode with a projection area smaller than that of the sample. Koray Palazoğlu et al. [24] compared RF (27.12 MHz) and microwave (915 MHz) tempering of frozen blocks of shrimp (1.75 kg) and observed that localized overheating occurred in samples subjected to microwave tempering. In contrast, RF tempering resulted in a relatively uniform overall temperature distribution. Palazoğlu and Miran [25] studied the effect of conveyor movement and sample vertical position on RF tempering of frozen lean beef (2.0 kg). It was demonstrated that tempering beef at the mid position in stationary mode and at high position on an upwardly inclined conveyor in continuous mode could achieve the best temperature uniformity. Li et al. [26] investigated the effects of shapes (cuboid, trapezoidal prism, step) and sizes (small: 160 × 102 × 60 mm^3^, medium: 220 × 140 × 60 mm^3^, large: 285 × 190 × 60 mm^3^) of frozen beef on RF tempering uniformity. It was reported that the worst heating uniformity occurred in the samples with sharp edges and steps (*STUI* 0.282) (*STUI*: simulated temperature uniformity index), followed by the trapezoidal prism (*STUI* 0.209) and cuboid shape samples (*STUI* 0.194). Additionally, heating uniformity increased as base area of the sample increased. Furthermore, Zhang et al. [27] studied the effects of different tempering methods and freeze-thaw cycles on melanosis and quality parameters of Pacific white shrimp. It was found that RF tempering effectively inhibited melanosis and reduced protein oxidation in Pacific white shrimp with its fast and uniform heating characteristics. Previous research explored not only the attributes of RF tempering but also methods to improve tempering uniformity on selected meat and fish products. However, not much research has been conducted on exploring the effectiveness of RF tempering of fish fillets and analyzing the fish quality. The lack of information hinders the application of RF tempering in the fish industry. A comprehensive comparison of the tempering processes among radio frequency tempering, water tempering and air tempering with an analysis of their tempering times, temperature distributions, and resulting fish quality attributes would provide new information to both researchers and industry people for accelerating the commercialization of RF tempering technology.

Dielectric properties are the essential factors affecting the electromagnetic wave absorption and energy conversion of materials [11]. Dielectric constant (*ε*′) and dielectric loss factor (*ε*″) are often used to characterize material dielectric properties [28]. The former represents the storage capacity of electromagnetic energy, while the latter reflects the ability of materials to absorb electromagnetic energy or convert electromagnetic energy into heat [29]. Furthermore, the dielectric properties of food are affected by many factors such as temperature and frequency, and also vary with food composition and quality. Many studies have been conducted using dielectric properties as a non-destructive method for predicting the quality and maturity of fruits and [30,31,32], eggs [33], milk [34] and meat maturity [35] and quality [36].

In this study, we selected salmon (*Salmo salar*) as a target fish for exploring the effectiveness of radio frequency tempering. The objective of this study was to compare RF tempering with water tempering and air tempering on the tempering rate, temperature distribution and physiochemical properties of salmon fillets. The investigated quality parameters included drip loss, cooking loss, color, water migration and texture properties.

## 2. Materials and Methods

### 2.1. Sample Preparation

Salmon fillets (*Salmo salar*), which were deep frozen in Norway, were purchased from a local supermarket in China (Lingang, Shanghai, China). The fillets were kept in a cooler filled with ice and transported to the laboratory within 0.5 h. The initial transport temperature of the salmon was −18 °C and there was no significant temperature fluctuation within the process. Salmon fillets were firstly cut and trimmed into cuboid shape (10 × 8 × 2.5 cm^3^), weighed to 240 ± 5 g, and then kept frozen at −20 °C for 24 h. After freezing, a *Φ*2 mm hole was drilled at the geometrical center of each fillet for temperature sensor insertion. The prepared samples were frozen at −20 °C for 24 h again before tempering experiments. Salmon fillets without freezing-thawing treatments were used as control samples for comparison. The initial moisture content of the sample was measured as 61.3% (AOAC 950.46).

### 2.2. Dielectric Properties

The dielectric properties of salmon were measured with an open-ended coaxial probe (Agilent N1501A, Agilent Technologies Inc., San Jose, CA, USA) connected with a network analyzer (Agilent E5071C, Agilent Technologies Inc., San Jose, CA, USA). The system was equipped with a custom-made cylindrical sample container (*d* = 2.5 cm, *h* = 10.0 cm) and an oil bath with temperature control device [37]. Before starting, the vector network analyzer was turned on firstly to preheat for 1 h, and the scanning type of the software was set to linear scanning with a scanning frequency range of 1 and 300 MHz. Probe calibration was conducted with air, short circuit and deionized water at 25.0 ± 0.5 °C. After that, the thawed salmon samples were minced by hand and stuffed into the sample container for measurement. Subsequent measurements were taken with a temperature range from −20 to 20 °C with a temperature interval of 5 °C by controlling the oil bath temperature. Each group of experiments was repeated three times. Detailed measuring procedures can be found in Chen et al. [38].

### 2.3. Penetration Depth

The penetration depth of an electromagnetic wave into a material refers to the vertical distance of the electromagnetic wave passing through the food when the intensity of the electromagnetic wave is reduced to 1/*e* (*e* = 2.71828) of the strength on the food surface. The equation used to determine penetration depth is defined as:(1)$$d_{p}=\frac{c}{2\sqrt{2}\pi f{\left[\varepsilon \prime \left(\sqrt{1+{\left(\frac{\varepsilon \prime \prime \;}{\varepsilon \prime }\right)}^{2}}-1\right)\right]}^{\frac{1}{2}}}$$
where *d_p_* is the penetration depth (m), c is the speed of light in free space (3 × 10^8^ m/s), *f* stands for the working frequency (MHz), *ε*′ represents the dielectric constant (F/m) and *ε*″ represents the dielectric loss factor (F/m).

### 2.4. Tempering Experiments

The prepared frozen salmon fillets were tempered from −20 to −4 °C with three methods: radio frequency tempering (RT), water immersion tempering (WT + 10 °C), and air tempering (AT + 10 °C); all tempering experiments were replicated three times in parallel.

#### 2.4.1. Radio Frequency Tempering (RT)

A 40.68 MHz, 400 W radio frequency oven (D20 Plus, Dotwil Intelligent Technology Co., Ltd., Shanghai, China) (Figure 1) with a fixed electrode gap of 122 mm was used for tempering experiment. A pre-calibrated fiber optic sensor (HQ-FTS-D1F00, Heqi guangdian Technology Co., Ltd., Xi’an, China) was inserted into the pre-drilled hole of one piece of salmon fillet for automatically monitoring and recording temperature histories during the tempering process. The sample was placed in the oven cavity for the tempering experiment. The experiment was stopped right away once the sample’s center temperature reached −4 °C. The treated sample was taken out of the oven immediately to obtain the top and bottom surface temperature distribution under a thermal infrared imager (FLIR A655sc, Wilsonville, OR, USA). The experiment was performed in triplicate with three different pieces of salmon fillets from different batches.

#### 2.4.2. Water Tempering (WT) and Air Tempering (AT)

Prepared frozen salmon fillets were placed in a thermostatic water bath (HH-S, Titan Technology Co., Ltd., Shanghai, China), and the water temperature was set to 10 ± 0.5 °C for the WT + 10 °C experiment. For the AT + 10 °C experiment, one piece of frozen salmon fillet was placed in atmospheric air at 10 ± 1 °C. Before the experiments started, thermocouple temperature sensors (OMEGA Engineering, Norwalk, CT, USA) were inserted into the centers of samples to monitor and record the temperature history until reaching −4 °C. After reaching the target temperature, the samples were taken out immediately to obtain the surface temperature picture using a thermal infrared imager (FLIR A655sc, Wilsonville, OR, USA). Each experiment with WT and AT was performed in triplicate with three different pieces of salmon fillets from different batches.

### 2.5. Drip Loss and Cooking Loss

The frozen samples were weighed before tempering experiments. After tempering, the salmon fillet samples were transferred to a 4 °C refrigerator for temperature equilibration for 4 h until fully thawed. After equilibration, samples were wiped dry with absorbent paper and weighed again. The drip loss was expressed as a percentage of water loss in the initial sample weight [39].
(2)$$\mathrm{Drip}\;\mathrm{loss}\;\left(\%\right)=\frac{weight\;of\;frozen\;sample\;\left(g\right)-weight\;of\;thawed\;sample\;\left(g\right)}{weight\;of\;frozen\;sample\;\left(g\right)}\times 100$$

Thawed salmon samples were cut into six cuboid pieces (3 × 2 × 1 cm^3^), weighed, and placed into polyethylene bags individually. The samples with bags were labeled and then immersed in a thermostatic water bath (HH-S, Titan Technology Co., Ltd., Shanghai, China) at 85 °C for 30 min. The salmon samples were weighed again after cooking, and the cooking loss was expressed as follow:(3)$$\mathrm{Cooking}\;\mathrm{loss}\;\left(\%\right)=\frac{weight\;of\;thawed\;sample\;\left(g\right)-weight\;of\;cooked\;sample\;\left(g\right)}{weight\;of\;uncooked\;sample\;\left(g\right)}\times 100$$

### 2.6. Color

The surface color of control and thawed samples were both measured using a handheld colorimeter (Minolta CR-400, Tokyo, Japan). Before measurements, the colorimeter was calibrated with a white standard calibration plate. The *L** (lightness), *a** (redness-greenness) and *b** (yellowness-blueness) values were obtained by contacting the probe to the surfaces of samples vertically. Each experiment was performed in triplicate at randomly selected locations on the sample surfaces.

### 2.7. Lipid Oxidation

The lipid oxidation degree of salmon fillets was determined by measuring the thiobarbituric acid reactive substances (TBARS) based on the methods of Zhu et al. [40] and Xuan et al. [41] with slight modifications. Around 10.0 g minced sample was placed in a conical flask and mixed with 50 mL 7.5% trichloroacetic acid with 0.1% EDTA. Then the mixture was shocked in a thermostatic water bath oscillator (SHZ-B, Titan Technology Co., Ltd., Shanghai, China) at a speed of 130 r/min for 30 min and then filtered. Five milliliter (5 mL) filtrate was transported into a glass test tube and mixed thoroughly with 5 mL 0.02 mol/L 2-thiobarbituric acid. The above tubes were placed in a water bath at 90 °C for 40 min, and then cooled down in 25 °C water for 30 min. After cooling down, 5 mL chloroform was added into the mixture solution, which was then shaken thoroughly by hand. Then, the supernatant was used to determine the absorbance value at 532 and 600 nm using a UV-Visible Spectrophotometer (Evolution 220, Thermo Fisher Scientific Inc., Waltham, MA, USA). Analyses were performed in *n* = 6 for each fish sample. The TBARS values were expressed in mg of malonaldehyde/kg of sample and calculated using the following equation:(4)$$\mathrm{TBARS}\;\left(\mathrm{mg}\;\mathrm{MDA}/\mathrm{kg}\right)=\frac{A_{532\;nm-A_{600\;nm}}}{155}\times \frac{1}{10}\times 72.6\times 1000$$

### 2.8. Protein Denaturation

Differential scanning calorimetry (DSC) is a common method for measuring the protein denaturation temperature and enthalpy to evaluate the protein denaturation degree of meat and fish products [42]. The experiment was conducted according to Rahbari et al. [43] with slight modifications. Around 10.0~15.0 mg samples were placed into a hermetically sealed aluminum pan, and thermal transition was assessed with a differential scanning calorimeter (Q2000, TA Instrument, New Castle, DE, USA). An empty pan was used as a reference. Samples were equilibrated at 20 °C for 2 min before heating at a rate of 5 °C/min till 100 °C. After each run, the maximum denaturation temperature (*T_max_*) and denaturation enthalpy (*ΔH*) were determined by analyzing the heat flow curves [44]. Experiments were performed for each treatment replicate.

### 2.9. Water Migration

Low-field nuclear magnetic resonance (LF-NMR) was used to analyze the water states and migration in salmon fillet samples. Each control and thawed sample was cut into 3 cubes (2 × 2 × 2 cm^3^) and wiped gently with absorbent paper. Each individual sample was placed in an NMR tube (*Φ* = 70 mm) at room temperature. *T*_2_ transverse relaxation measurements were performed using an LF-NMR analyzer (MesoMR23-060H.I, Niumag Electronic Technology Co., Ltd., Shanghai, China) with a magnetic field strength of 0.4 T corresponding to a proton resonance frequency of 21 MHz. *T*_2_ was measured at 32 °C using the Carr-Purcell-Meiboom-Gill (CPMG) pulse sequence [45,46]. The *T*_2_ measurements were conducted with a time delay between the 90° and 180° pulse (τ) of 150 µs. For each sample, 16 scans were acquired at 2 s intervals with 8000 echoes. Detailed experimental procedure can be found in Aursand et al. [47]. Each measurement was performed in triplicate.

### 2.10. Microstructure

Tissues of salmon fillet sample were cut into 3 cubes (5 × 5 × 5 mm^3^) and fixed at 4 °C for 24 h in 10% phosphate-buffered (pH 6.9~7.1) formaldehyde and dehydrated with a gradient series of ethanol solutions, and then embedded in paraffin wax. Samples were then cut into 5 µm thick slices and dried for 24 h, then dewaxed and stained with hematoxylin and eosin [48]. The specimens were observed and photographed by a light microscope (BZ-9000, Keyence, Osaka, Japan). Three measurements were carried out for each treatment.

### 2.11. Statistical Analysis

Three independent trials were performed to test the effects of tempering methods on physicochemical properties of salmon, including drip loss, cooking loss, TBARS, denaturation temperatures and enthalpy, color, and water migration. One-way analysis of variance (ANOVA) was used to analyze the previously mentioned physiochemical parameters following a Duncan’s multiple range test, expressed as mean ± standard deviation (SD) (SPSS 25.0, Chicago, IL, USA). Significant difference (*p* < 0.05) was used to compare treatment means. All figures were plotted with OriginPro 9.0 (OriginLab Co., Northampton, MA, USA).

## 3. Results and Discussions

### 3.1. Dielectric Properties

Table 1 shows that the dielectric constant and dielectric loss factor of salmon increase with temperatures within the range of −20 to 20 °C. At all selected frequencies, the dielectric constant and dielectric loss factor increased slowly from −20 to −10 °C, and then increased rapidly from −10 to −5 °C due to the melting of ice crystals (Figure 2). In general, both dielectric constant and loss factor increased as temperature increased from −20 to 20 °C which was attributed to the increasing water molecule dipole rotation and ionic conductivity [49]. This would possibly result in significant temperature non-uniformity at the sample edges, since the edges usually tend to absorb more energy and the localized heating results in rapid melting of ice crystals, which further aggravates the non-uniform heating. Table 1 also lists the penetration depth of salmon at selected frequencies and temperatures between −20 and 20 °C. The penetration depth decreased with increasing temperature at three different frequencies, and the lower the frequency was, the greater the decrease was. The tendency was similar to those reported in literature, namely that the penetration depth decreases as the temperature increases for most foods [36]. In addition, the penetration depth of the sample at 13.56 MHz was the largest, followed by 27.12 and 40.68 MHz, which is consistent with the results of some researchers who considered that the penetration depth usually decreases with increases in frequency [50,51]. According to the results, it can be predicted that the thawing uniformity of the sample will be better at lower frequency.

### 3.2. Tempering Rate

The tempering rates of salmon fillets with different tempering methods are shown in Figure 3. The tempering rate of RT was higher than that of AT + 10 °C and WT + 10 °C, and the tempering time of WT (15 min) and AT (64 min) were 3 and 12.8 times longer than RT (5 min), respectively. Choi et al. [52] also reported that the tempering rate of RT (at 400 W) of pork loin (100 × 100 × 70 mm^3^) was 100 times faster than that of AT. The reason is that WT and AT elevate the sample temperature mainly by heat convection and conduction, and the heating rate relies on the limited convective heat transfer coefficient and thermal conductivity, while radio frequency generates heat within the food samples rapidly and volumetrically [53,54]. Meanwhile, the tempering rate from −20 to −5 °C was much faster than from −5 to −4 °C. This is because the thermal conductivity of ice is four times that of water [55], and the immobilized water started to shift to the mobilized state when the temperature reached −5 °C, causing the slow rise in temperature.

### 3.3. Surface Temperature Distribution

The temperature distribution on the top and bottom surfaces of salmon samples after different tempering methods are shown in Figure 4. For RT, the temperature distribution on the top surfaces of salmon fillets were more uniform than for the other two tempering methods, and the temperature over most of the area was controlled below 0 °C. The result was similar to the results reported by Zhang et al. [27]. However, overheating at the sample edges and corners was observed since edges and corners are a convergence of many surfaces, and the electromagnetic field intensity at these locations is higher than in the rest of the area, thus resulting in more severe heating [26,56]. However, the bottom surfaces showed WT and AT produced more uniform heating, and RT showed a much higher temperature than on the top surfaces. This is possibly because water migration occurred during the tempering process and free water transferred to the bottom surface, resulting in higher a heating rate on the bottom surface.

From Table 2, the maximum temperature on the surfaces of RT, WT + 10 °C and AT + 10 °C samples were 7.3 °C, 11.8 °C, 7.2 °C and 13.4 °C, 8.4 °C, 2.2 °C for the top and bottom surfaces, respectively. The smallest differences in surface temperature were found in AT + 10 °C, and also the standard deviation of AT + 10 °C was the smallest, indicating that AT + 10 °C was the most uniform tempering method. Similar results were reported by Zhu et al. [40]. This is because the convective heat transfer coefficient between water and the salmon fillet is larger than that between air and the [57], resulting in a faster temperature increase but worse tempering uniformity than with air tempering. Combining the heating uniformity results with tempering rates, it was noted that although AT had the best tempering uniformity, its tempering rate was the lowest. RT had the highest tempering rate, but the tempering uniformity could be better controlled before extending it to industrial use by reducing the edge effect and absorbing the drip loss during tempering. In most industrial radio frequency equipment, the heating rate can be further regulated by varying the electrode gap, which also influences the heating uniformity.

### 3.4. Drip and Cooking Loss

The drip and cooking loss of salmon fillet under different tempering conditions are presented in Table 3. The drip and cooking loss usually contain water and water-soluble substances. A lower drip and cooking loss represent a higher water holding capacity of raw and cooked meat, respectively. It could be observed that the tempering methods had a significant effect on both drip and cooking loss (*p* < 0.05). For tempering loss, AT + 10 °C produced the least amount (0.43%) of drip loss while WT + 10 °C produced the most (1.03%). Meanwhile, RT, WT and AT resulted in a cooking loss of 13.58%, 15.75% and 12.26%, respectively, after a freezing and tempering cycle, while the control sample exhibited a cooking loss of 11.34%. The higher drip and cooking loss of WT + 10 °C samples were also possibly a result of increased myosin denaturation [58]. Xia et al. [58] demonstrated that freezing and thawing cycles decreased the stability of myosin and actin, changed the microstructure of myofibrillar protein, and finally affected the physical attributes of meat, such as juiciness and texture. Damaged structure and denatured proteins reduced the ability of muscle to retain water, resulting in a lower water holding capacity [8]. This result agreed with Farag et al. [21], who observed that air tempering resulted in higher drip loss (18.0%) than RT (9.0%) in whole lean beef samples. The results were in accordance with the temperature distribution in the fish fillets, where WT samples had the worst tempering uniformity and resulted in the highest amount of drip and cooking loss. This indicates that temperature is a key factor affecting drip and cooking loss.

### 3.5. Color

Studies have shown that consumers usually relate flesh color with the freshness and quality of fish [59]. Table 4 shows the color (*L**, *a**, *b**and *ΔE* values) of salmon fillets before and after different tempering methods. The total color difference parameter (*ΔE*) showed that treated salmon fillets had significant differences with the control sample. The AT + 10 °C sample differed the least from the control sample compared with the WT + 10 °C and RT samples. The *L** value of thawed samples increased significantly (*p* < 0.05) compared to the control sample, and the highest increase in the *L** value was discovered in the WT + 10 °C sample, while the lowest was observed in the AT + 10 °C sample. It was discovered that all tempering methods had significant (*p* < 0.05) effects on the lightness (*L**) of salmon fillets, but the WT + 10 °C and AT + 10 °C methods showed the most and least negative effect on the lightness (*L**) of salmon sample, respectively. Because of the long tempering time and the small temperature differences between salmon fillet and air during the final tempering stage, the surface of fillet flesh was protected from negative change. Additionally, the higher tempering rate and relatively uniform tempering of RT resulted in a tempered sample color close to that of the original salmon fillet.

As for the *a** value, no significant (*p* > 0.05) difference in redness (*a**) was found between the control sample and the samples treated with three different tempering methods. The reason is possibly that the dominant pigment in salmon fillet is astaxanthin, which is fat-soluble and not easily lost with drip loss. However, the redness of the sample treated with AT + 10 °C increased, which may be because the low temperature resulted in less oxidation reaction. For the *b** value, no significant difference (*p* > 0.05) was found between the WT + 10 °C sample and control sample, and AT + 10 °C and RT had a lower *b** value. The change in yellowness of salmon fillet was possibly due to protein and lipid oxidation [22].

### 3.6. TBARS

The lipid oxidation degree was quantified by the TBARS values of the fish samples. The higher the TBARS value, the higher the degree of lipid oxidation. The TBARS values of the control and thawed salmon are shown in Figure 5. It could be seen that the TBARS values increased significantly after all tempering processes (*p* < 0.05) except for AT + 10 °C. Ke et al. [60] suggested that levels below 8 mg MDA/kg fish flesh could be indicative of good quality. The TBARS value of control salmon was 1.027 mg MDA/kg, while that of WT + 10 °C was the highest (3.102 mg MDA/kg), followed by RT (2.135 mg MDA/kg) and AT + 10 °C (1.054 mg MDA/kg). This result was similar to that of Xia et al. [39], who reported that AT and WT resulted in TBARS value increases of 56.2% and 71.1%, respectively, compared to fresh longissimus muscle flesh. RT fish samples showed a higher tempering rate (Figure 3) and less lipid oxidation than WT samples (Figure 5). Comparatively, WT samples were exposed directly to water, and the differences in pressure between water and the samples caused an oxidation reaction and damage to the samples’ microstructure.

### 3.7. Protein Denaturation

Three heat flow peaks were observed in salmon fillets representing myosin, sarcoplasmic proteins and actin. The peak temperature and endothermic values, *T_max_* and *ΔH*, were the denaturation temperatures and enthalpy of myofibrillar protein, respectively. The values of *T_max_* and *ΔH* reflect the degree of protein denaturation compared to that of the control sample. The smaller the *T_max_* and *ΔH* values, the higher the degree of protein denaturation [61].

As shown in Table 5, no significant differences (*p* > 0.05) were observed among control sample and thawed samples in their *T*_2*max*_, *T*_3*max*_ *ΔH*_2_, and *ΔH*_3_ values. However, for *T*_1*max*_ and ΔH_1_, a significant decrease (*p* < 0.05) in peak temperature was found in the WT + 10 °C and RT samples, indicating these two tempering methods had a stronger negative effect on myosin than AT + 10 °C. This result illustrated that among the three tempering methods, AT + 10 °C was the most appropriate tempering method for protein preservation, followed by RT and WT + 10 °C. Further, it was found that myosin was more easily damaged and denatured in freeze-temper cycles. The denaturation of myofibrillar protein was also associated with the lipid oxidation and microstructure damage [48].

### 3.8. Moisture Migration

Low-field nuclear magnetic resonance (LF-NMR) was used to study the water migration mobility in food materials [62], which was expressed as transverse relaxation time, *T*_2_. A longer *T*_2_ indicates a stronger mobility of water. As shown in Figure 6, there were three peaks in salmon fillets corresponding to *T*_2_ of the three states of water in the samples: *T*_21_ (0–10 ms), representing bound water with the strongest binding force among three states; *T*_22_ (10–100 ms), representing immobilized water or intracellular water that was entrapped in the myofibrillar network and was usually the predominant water component in flesh; and *T*_23_ (100–1000 ms), representing free water that was restricted by capillary forces and could be lost by heating and mechanical damage [63].

From Figure 6, it can be seen that the changes in bound water (*T*_21_) in all samples were subtle. This is possibly because the bound water was tightly bound to proteins and other macromolecular substances and was barely affected by the tempering processes. It was observed that WT + 10 °C led to longer *T*_21_ and *T*_22_ and larger amplitudes compared to the control sample and the other two treated samples, which revealed that the bound and immobile water shifted to free water. The *P*_21_, *P*_22_ and *P*_23_ of fresh sample were 2.9%, 97.0% and 0.1%, respectively (Figure 7). There were significant differences (*p* < 0.05) in the peak area (*P*_21_, *P*_22_ and *P*_23_) of samples after different tempering treatments. After tempering, a significant decrease in *P*_22_ (the peak area of *T*_22_, immobile water) was observed in the sample treated with WT + 10 °C, followed by that of RT and AT + 10 °C. The increase in *P*_23_ of the WT + 10 °C sample indicated a possible migration from tightly bound (*T*_22_) to loosely bound (*T*_23_) water as a result of microstructure damage and protein denaturation. McDonnell et al. [63] also found that WT had a significant destructive effect on pork, resulting in severe myofibrillar protein denaturation.

### 3.9. Microstructure

The microstructures of the control and thawed tissue tempered with different methods are shown in Figure 8. After tempering, myofibrils with clear boundaries and a larger intercellular space were observed in tempered samples, especially the WT + 10 °C sample. From the severely deformed myofibrils observed in the WT + 10 °C sample, it could be speculated that the sample tissue suffered significant mechanical injury from water tempering and released the moisture and cellular substances to the intercellular spaces or as tempering loss. The microstructures of RT and AT + 10 °C samples were found to be less damaged since the spaces between the muscle fiber bundles were smaller than those of WT + 10 °C sample. This was consistent with the results of tempering loss, cooking loss and moisture migration of salmon flesh. RF tempering shortened the tempering time and reduced the damage to muscle fiber and fragmentation of cell structures [10]. Zhang et al. [64] also reported that improper tempering methods resulted in protein denaturation and structural damage to muscle tissue and cells.

## 4. Conclusions

The tempering rate, temperature distribution and physiochemical properties of frozen salmon fillets subjected to radio frequency (RT), water (WT + 10 °C) and air (AT + 10 °C) tempering were evaluated. RT presented the shortest tempering time, less tempering and cooking loss, and a lower degree of lipid oxidation and protein denaturation compared to WT + 10 °C. These results were consistent with the water mobility results from LF-NMR and muscle histology. Although WT + 10 °C produced a faster tempering rate than AT + 10 °C, it was an undesirable method for salmon fillet tempering because the physiochemical properties were not better than those obtained with AT + 10 °C. In conclusion, the best overall quality was found in the salmon fillet treated with AT + 10 °C, but with the drawback of long tempering time. RT is a promising technology and has been proposed as an alternative novel salmon tempering method for the food industry, offering fast and relatively uniform tempering with a higher quality retention rate. RT throughput can be further enhanced with continuous processing.

This study provided a general evaluation of salmon fillet quality after various tempering methods, and showed that RF is a better method of thawing frozen salmon fillets. However, industrial-scale tempering of frozen salmon fillets in bulk or of whole salmon fish has not been studied yet. Thus, the industrial processing parameters for fishery product tempering with RF should be further explored, including analyzing the influencing factors that contribute to tempering uniformity, quality and energy consumption. Furthermore, whole fish and fish fillets could present significantly different effects in parallel-plate RF tempering and will require further investigation of the influences on process development from their complex shapes and composition. In addition, the endpoint tempering temperature should be optimized on the basis of salmon fish processing procedures.

## Figures and Tables

**Figure 1 foods-11-00893-f001:**
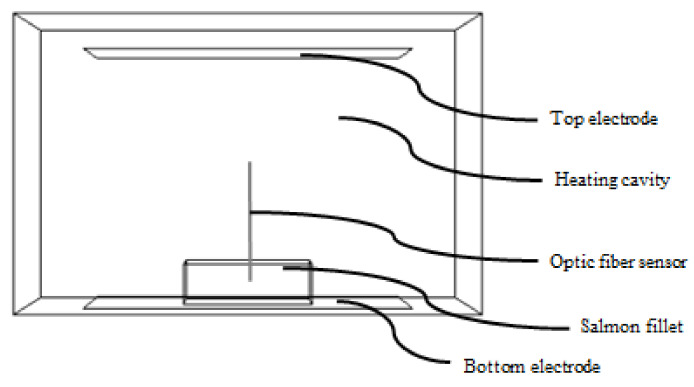
Layout of RF system and placement of salmon fillet sample and fiber optic sensor.

**Figure 2 foods-11-00893-f002:**
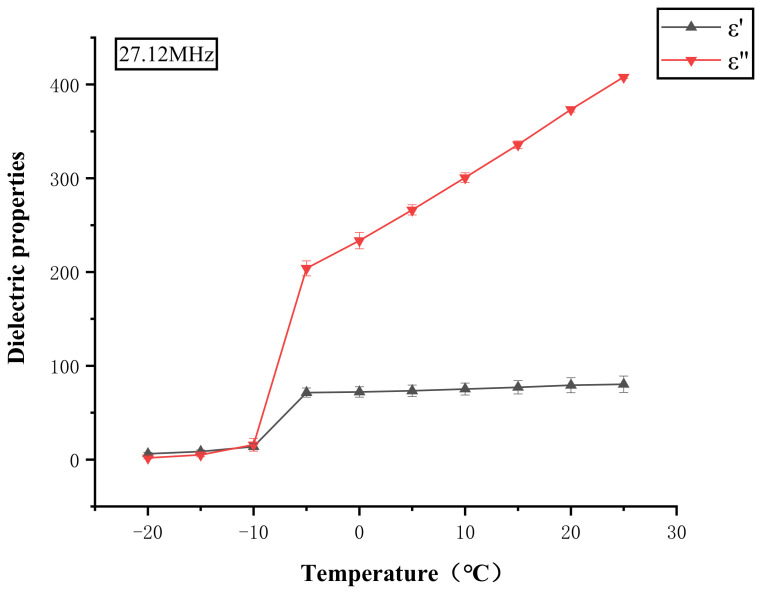
Dielectric constant (*ε*′) and dielectric loss factor (*ε*″) of salmon as a function of temperature at 27.12 MHz.

**Figure 3 foods-11-00893-f003:**
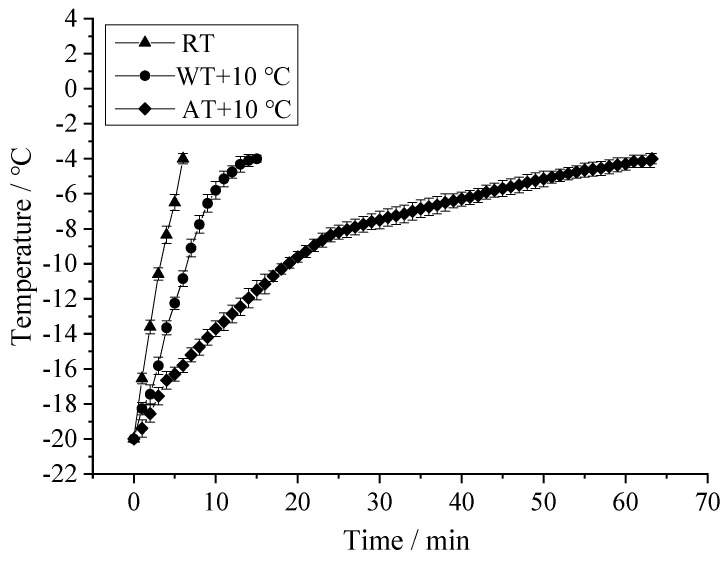
Temperature–time histories of salmon samples during different tempering treatments. (RT: Radio frequency tempering; WT + 10 °C: Water tempering at 10 ± 0.5 °C; AT + 10 °C: Air tempering at 10 ± 1 °C).

**Figure 4 foods-11-00893-f004:**
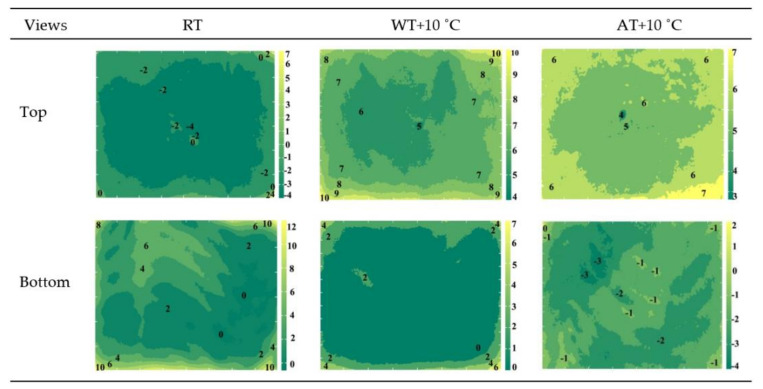
Surface temperature distribution of salmon fillets after different tempering treatments when the sample center reached −4 °C. (RT: Radio frequency tempering; WT + 10 °C: Water tempering at 10 ± 0.5 °C; AT + 10 °C: Air tempering at 10 ± 1 °C).

**Figure 5 foods-11-00893-f005:**
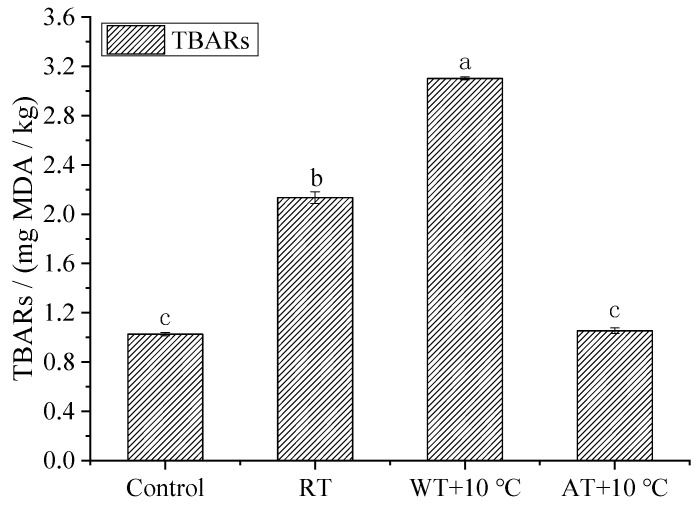
Effects of different tempering methods on TBARS of salmon. (RT: Radio frequency tempering; WT + 10 °C: Water tempering at 10 ± 0.5 °C; AT + 10 °C: Air tempering at 10 ± 1 °C). Different letters on the bar (a,b,c) indicate significant differences (*p* < 0.05) among samples treated with different methods.

**Figure 6 foods-11-00893-f006:**
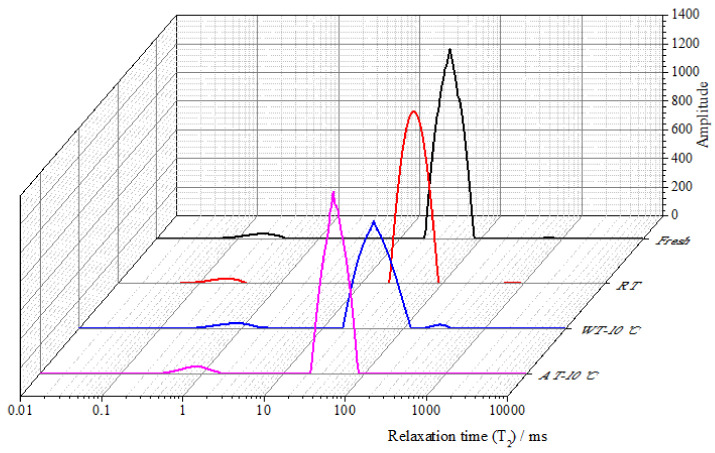
Transverse relaxation time (*T*_2_) curves of salmon samples after tempering, evaluated by LF-MNR. (RT: Radio frequency tempering; WT + 10 °C: Water tempering at 10 ± 0.5 °C; AT + 10 °C: Air tempering at 10 ± 1 °C).

**Figure 7 foods-11-00893-f007:**
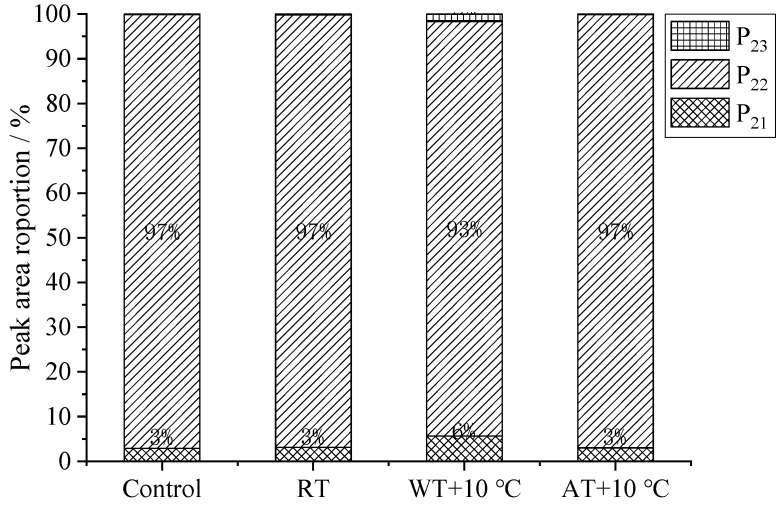
The relaxation time (*T*_2_) corresponding to relative peak areas (*P*_21_, *P*_22_ and *P*_23_) of salmon fillets after different tempering treatments. (RT: Radio frequency tempering; WT + 10 °C: Water tempering at 10 ± 0.5 °C; AT + 10 °C: Air tempering at 10 ± 1 °C).

**Figure 8 foods-11-00893-f008:**
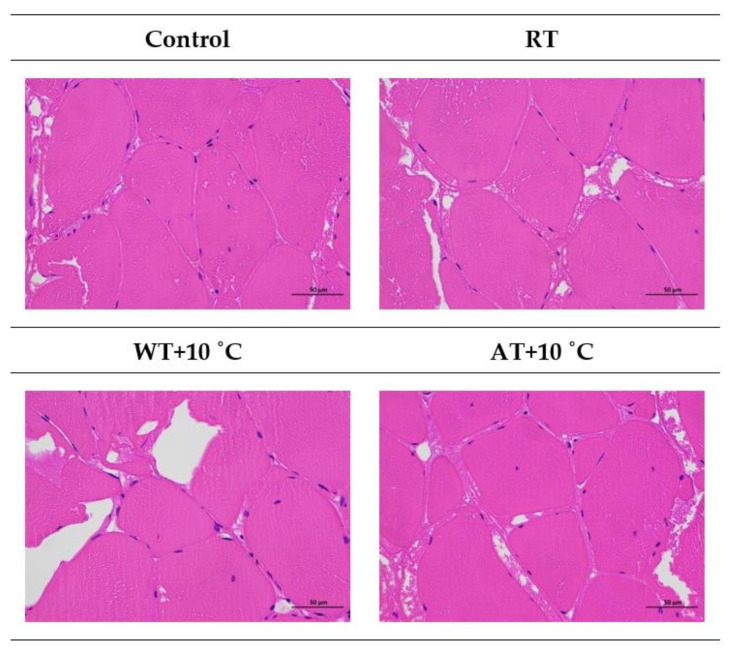
Microstructure of salmon fillet tissues after different tempering methods under light microscope (magnification ×400). (RT: Radio frequency tempering; WT + 10 °C: Water tempering at 10 ± 0.5 °C; AT + 10 °C: Air tempering at 10 ± 1 °C).

**Table 1 foods-11-00893-t001:** Dielectric properties and penetration depth (*dp*) of salmon at 13.56, 27.12, and 40.68 MHz and temperatures between −20 and 20 °C.

T (°C)	Parameters	13.56 MHz	27.12 MHz	40.68 MHz
−20	ε′	11.25 ± 0.31 ^Ad^	7.54 ± 0.02 ^Bb^	2.64 ± 0.03 ^Cc^
	ε″	6.10 ± 0.17 ^Af^	2.82 ± 0.09 ^Bh^	1.65 ± 0.04 ^Ch^
	dp/cm	200.41 ^Aa^	174.53 ^Ba^	120.56 ^Ca^
−15	ε′	15.39 ± 0.06 ^Ad^	10.16 ± 0.02 ^Bb^	4.79 ± 0.11 ^Cbc^
	ε″	14.17 ± 0.26 ^Af^	6.81 ± 0.12 ^Bh^	3.24 ± 0.56 ^Ch^
	dp/cm	105.97 ^Ab^	86.56 ^Bb^	83.45 ^Bb^
−10	ε′	24.13 ± 0.08 ^Ad^	13.76 ± 3.23 ^Bb^	11.41 ± 3.45 ^Bb^
	ε”	45.05 ± 4.37 ^Af^	15.74 ± 6.78 ^Bg^	9.93 ± 5.09 ^Bg^
	dp/cm	47.96 ^Ac^	46.58 ^ABc^	43.09 ^Bc^
−5	ε′	97.36 ± 5.99 ^Ac^	71.42 ± 4.88 ^Ba^	63.91 ± 4.04 ^Ba^
	ε″	432.35 ± 23.38 ^Ae^	203.89 ± 8.03 ^Bf^	140.43 ± 4.94 ^Cf^
	dp/cm	13.40 ^Ad^	10.36 ^ABd^	8.73 ^Bd^
0	ε′	101.15 ± 8.35 ^Abc^	72.07 ± 5.54 ^Ba^	64.55 ± 4.52 ^Ba^
	ε″	498.56 ± 27.76 ^Ade^	233.50 ± 8.69 ^Be^	159.66 ± 5.30 ^Ce^
	dp/cm	12.34 ^Ad^	9.49 ^ABd^	8.00 ^Bd^
5	ε′	104.74 ± 9.50 ^Abc^	73.43 ± 6.08 ^Ba^	65.69 ± 4.72 ^Ba^
	ε″	572.69 ± 45.01 ^Acd^	266.26 ± 5.47 ^Bd^	182.33 ± 2.95 ^Cd^
	dp/cm	11.40 ^Ad^	8.75 ^Ad^	7.34 ^Ad^
10	ε′	111.48 ± 9.21 ^Aabc^	75.22 ± 6.37 ^Ba^	66.91 ± 4.57 ^Ba^
	ε″	648.70 ± 53.84 ^Abc^	300.79 ± 5.09 ^Bc^	205.87 ± 2.40 ^Cc^
	dp/cm	10.65 ^Ad^	8.13 ^Ad^	6.79 ^Ad^
15	ε′	115.91 ± 11.57 ^Aab^	77.10 ± 7.08 ^Ba^	68.21 ± 4.92 ^Ba^
	ε″	722.13 ± 61.82 ^Aab^	335.76 ± 3.99 ^Bb^	229.94 ± 1.74 ^Cb^
	dp/cm	10.04 ^Ad^	7.62 ^Ad^	6.34 ^Ad^
20	ε′	122.97 ± 13.03 ^Aa^	79.27 ± 8.01 ^Ba^	69.65 ± 5.44 ^Ba^
	ε″	804.56 ± 74.99 ^Aa^	373.25 ± 2.63 ^Ba^	256.32 ± 0.80 ^Ca^
	dp/cm	9.48 ^Ad^	7.16 ^Ad^	5.93 ^Ad^

Different lowercase letters in the same column indicate significant differences (*p* < 0.05). Different uppercase letters in the same row indicate significant differences (*p* < 0.05). ε′: dielectric constant, ε″: dielectric loss factor, dp: penetration depth.

**Table 2 foods-11-00893-t002:** Top and bottom surface temperature of salmon after different tempering treatments. (RT: Radio frequency tempering; WT + 10 °C: Water tempering at 10 ± 0.5 °C; AT + 10 °C: Air tempering at 10 ± 1 °C).

Treatments	Views	Maximum Temperature/(°C)	Minimum Temperature/(°C)	Standard Deviation/(°C)
RT	top	7.3	−5.2	±1.1
	bottom	13.4	−0.6	±2.2
WT + 10 °C	top	11.8	3.7	±0.9
	bottom	8.4	−0.5	±1.1
AT + 10 °C	top	7.2	3.3	±0.5
	bottom	2.2	−3.9	±0.6

**Table 3 foods-11-00893-t003:** Effects of different tempering methods on drip loss and cooking loss of salmon. (RT: Radio frequency tempering; WT + 10 °C: Water tempering at 10 ± 0.5 °C; AT + 10 °C: Air tempering at 10 ± 1 °C).

Treatments	Drip Loss/(%)	Cooking Loss/(%)
Control	-	11.34 ± 0.02 ^b^
RT	0.66 ± 0.052 ^b^	13.58 ± 0.63 ^ab^
WT + 10 °C	1.029 ± 0.050 ^a^	15.75 ± 0.06 ^a^
AT + 10 °C	0.43 ± 0.021 ^b^	12.26 ± 0.48 ^b^

The superscript of the numbers (a,b) indicate significant differences (*p* < 0.05) between samples treated with different methods.

**Table 4 foods-11-00893-t004:** Effects of different tempering methods on the color of salmon flesh. (RT: Radio frequency tempering; WT + 10 °C: Water tempering at 10 ± 0.5 °C; AT + 10 °C: Air tempering at 10 ± 1 °C; *L**: lightness; *a**: redness-greenness; *b**: yellowness-blueness; *ΔE*: the total color difference).

Treatments	*L**	*a**	*b**	*ΔE*
Control	42.27 ± 0.48 ^b^	14.29 ± 0.33 ^a^	13.88 ± 0.32 ^a^	-
RT	44.46 ± 0.94 ^a^	13.57 ± 0.83 ^a^	12.14 ± 0.85 ^b^	2.89
WT + 10 °C	44.74 ± 0.32 ^a^	13.35 ± 0.73 ^a^	12.80 ± 0.52 ^ab^	2.85
AT + 10 °C	43.34 ± 1.00 ^a^	14.41 ± 0.23 ^a^	12.10 ± 0.60 ^b^	2.08

The superscript of the numbers (a,b) indicate significant differences (*p* < 0.05) between samples treated with different methods.

**Table 5 foods-11-00893-t005:** Effects of different tempering methods on *T_max_* and *ΔH* (the denaturation temperature and enthalpy of myofibrillar protein, respectively) of salmon. (RT: Radio frequency tempering; WT + 10 °C: Water tempering at 10 ± 0.5 °C; AT + 10 °C: Air tempering at 10 ± 1 °C).

Treatment	*T*_1*max*_/°C	*ΔH*_1_/(J/g)	*T*_2*max*_/°C	*ΔH*_2_/(J/g)	*T*_3*max*_/°C	*ΔH*_3_/(J/g)
Control	58.18 ± 0.82 ^a^	0.0779 ± 0.002 ^a^	67.88 ± 0.38 ^a^	0.0962 ± 0.001 ^a^	77.27 ± 0.06 ^a^	0.2315 ± 0.003 ^a^
RT	57.24 ± 0.36 ^b^	0.0662 ± 0.004 ^b^	67.51 ± 0.23 ^a^	0.0928 ± 0.001 ^ab^	76.56 ± 0.12 ^a^	0.2309 ± 0.002 ^a^
WT + 10 °C	56.93 ± 0.14 ^b^	0.0623 ± 0.005 ^b^	67.04 ± 0.19 ^a^	0.0890 ± 0.003 ^b^	76.55 ± 0.23 ^a^	0.2200 ± 0.006 ^a^
AT + 10 °C	57.88 ± 0.17 ^a^	0.0718 ± 0.003 ^a^	67.75 ± 0.21 ^a^	0.0941 ± 0.002 ^ab^	77.10 ± 0.24 ^a^	0.2307 ± 0.004 ^a^

The superscripts on the numbers (a,b) indicate significant differences (*p* < 0.05) between samples treated with different methods.

## Data Availability

Not applicable.

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
