# Peer review of "Effects of Radio Frequency Tempering on the Temperature Distribution and Physiochemical Properties of Salmon (Salmo salar)"

_foods, 2022, doi:10.3390/foods11060893_

Round 1

Reviewer 1 Report

Effects of radio frequency tempering on the temperature distribution and physiochemical properties of salmon (Salmo salar)

In general, the paper has a friendly writing, easy to read, the figures and tables are also simple, there are no major objections. However, the labeling WT-10 ˚C and AT-10 ˚C suggests that the temperature to be reached is -10°C, please correct this.

It is suggested to change the nomenclature of the treatments AT-10°C, WT-10°C by AT+10°C WT+10°C, so as not to confuse the results and the graph.

Abstract:

Line 8-9: correct this sentence please, it sounds like fish is less perishable before cooking, and more perishable when cooked.

It would be interesting to include in the abstract some conclusion of the work

Introduction:

Lines 42-43: Use more references to justify these intensities (since they claim to be common).

Line 92: please correct this sentence

Materials and methods:

Sample preparation

At what temperature were the fillets purchased?

What change in temperature did they have during the transfer?

 line 107, please indicate the initial transport temperature of the salmon. If there were significant temperature changes, because your research deals with tempering. Indicate if the samples presented exudation

DSC

 Line 206 : Is it DSC disk or pans?

Microestructure

Line 232: fix parentheses

Results:

Line 279: please specify Li 2019, identify as a and b in each citation each time they are used.

Tempering rate

Lines 289-271: add reference to this sentence

Surface temperature distribution

Lines 308-311: put references that support these sentences.

Color

It is suggested to include values ​​of ΔE or Δa* in the table, since it can provide a clear general discussion on the color changes of salar salmon and specifically on the difference of pink or red color with Δa*. (Attach equations in methodology)

Por qué no se calculó ∆E u otros parámetros con las coordenadas L*, a* y b*?

Why weren't ∆E or other parameters calculated with the coordinates L*, a* and b*?

 Line 280, correct the temperature unit -5 ̊˚C, to -5°C.

Please correct the references; there are references not cited in the manuscript.

Suggested to include recent literature referring to RF such as (Zhang et al., 2021, Choi et al., 2017, Cai et al., 2019)

In point 2.4.1 Radiofrequency tempering (RT), justify electrode gap and radiofrequency power. (Previous analysis, research by other authors, or are the default characteristics of the equipment).

On lines 329 to 331, mention the authors' research on myosin denaturation.

TBARS

Are the results within the range of freshness suitable for human consumption in accordance with any regulation?

Is there a relationship between these results and the time it took to reach the target temperature?

In section 2.3. Penetration depth, describe the following variables of the equation: f, ε', ε" and the corresponding units.

Author Response

We would like to thank you for the detailed comments and suggestions helping us to improve the quality of our work. We have addressed the questions specifically in the response, please check the file and let us know if there is any further question. Thanks again for your help!

Reviewer 2 Report

The manuscript has compared three different methods of thawing and tempering of frozen salmon. Adequate data have been presented and discussed. However, the manuscript could be improved as follow:

  • Some sentences must be rewritten for better understanding. For example, line 245: Table 1 and Fig. 2…. Should be checked. The manuscript should be checked for English grammar as well.
  • Table 1: significant differences should be indicated using lowercase and uppercase letters.
  • Line 260: further discussion about temperature is necessary. There are some differences for dielectric constant (ε') and dielectric loss factor (ε") in different temperatures which could be discussed. For instance, as it has been explained and lower frequency has been preferred in line 260-261, this could also be discussed for temperature. It is better to analyze Table 1 using two-way ANOVA.
  • Table 2: standard deviations have not been written correctly. It should be checked and amended.
  • Line 375: “RT resulted in a higher tempering rate, little…” from Fig 5 it could not be concluded.

Author Response

(The authors gave the same response as above.)

Reviewer 3 Report

Reviewer comments:

  1. Abstract: What kind of tempering process we are dealing with? Be specific. The RF isn’t anywhere seen in the abstract of what was actually happening to the salmon meat as if it exposed for specific process (e.g. cold sterilization etc.). What is your control sample? The DOE was quite straightforward. Any specific reason of choosing those range? What kind of physicochemical properties we were dealing with? Be specific.
  2. Method: Do you check any of the MC of the samples prior to treatment? It could be that the MC might have affected the treatment too. What is your control samples? Different batches were used why? Any specific reason? The DOE was quite vague.
  3. If microstructure was considered in the analysis, the mechanical properties of the meaty structure should be considered as well. Any results of the toughness of the meaty part? It is heavily related to the microstructure.
  4. Figure 5: You did include the control sample of the salmon (fresh). Define fresh of the sample used? Don’t see any of the fresh sample mention elsewhere.
  5. Figure 4: Any compositional coloration taken for the fresh samples? This has to be included elsewhere consistently.

Author Response

(The authors gave the same response as above.)

Round 2

Reviewer 3 Report

The authors have answered all the questions needed. Thank you